# Operational Meaning of Classical Fidelity and Path Length in Kubo–Mori–Bogoliubov Fisher Geometry

**DOI:** 10.3390/e27010042

**Published:** 2025-01-07

**Authors:** Lajos Diósi

**Affiliations:** 1Wigner Research Center for Physics, P.O. Box 49, H-1525 Budapest 114, Hungary; diosi.lajos@wigner.hu; 2Department of Physics of Complex Systems, Eötvös Loránd University, Pázmány Péter stny. 1/A, H-1117 Budapest, Hungary

**Keywords:** Bhattacharyya coefficient, fidelity, Fisher–Bures metric, quantum state transport, entropy production

## Abstract

We show that the minimum entropy production in near-reversible quantum state transport along a path is a simple function of the path length measured according to the Fisher–KMB metrics. Hence, for the sharp values of path lengths, also called statistical lengths, we obtain the operational meaning to quantify the residual irreversibility in near-reversible state transport. In the classical limit, the Bhattacharyya fidelity is found to have a sharp operational meaning after eighty years.

## 1. Introduction

A quantitative comparison between two probability distributions, $p_{\alpha }$ and $q_{\alpha }$, can be based on the Bhattacharyya coefficient published in 1945 [1]:(1)$$F\left(p,q\right)=\sum_{\alpha }\sqrt{p_{\alpha }q_{\alpha }}\;\in \left[0,1\right].$$
This function serves as a measure F∈[0,1] of closeness between *p* and *q*. Fifty years after the publication of the Bhattacharyya coefficient, to measure the closeness of two quantum states, $\hat{\rho }$ and $\hat{\sigma }$, Jozsa borrowed an expression known in mathematics [2,3] and called it fidelity [4]:(2)$$F\left(\hat{\rho },\hat{\sigma }\right)={∥\sqrt{\hat{\rho }}\sqrt{\hat{\sigma }}∥}_{\mathrm{tr}}=\mathrm{tr}\sqrt{\sqrt{\hat{\sigma }}\hat{\rho }\sqrt{\hat{\sigma }}}\;\in \left[0,1\right].$$
Quantum fidelity is the central tool in modern quantum informatics [5]. In the special (classical) case when $[\hat{\rho },\hat{\sigma }]=0$, we can write $\hat{\rho }=\mathrm{diag}\left[p_{1},p_{2},\ldots ,p_{\alpha },\ldots \right]$ and $\hat{\sigma }=\mathrm{diag}\left[q_{1},q_{2},\ldots ,q_{\alpha },\ldots \right]$. Quantum fidelity (Equation 2) coincides with classical fidelity (Equation 1).

One of our goals is the operational interpretation of F(q,p) because it is not known [5]. For quantum fidelity $F(\hat{\rho },\hat{\sigma })$, a simple operational interpretation has already been proposed [6], based on Uhlmann’s theorem [3], and it is as follows: *Assume that $\hat{\rho }$ and $\hat{\sigma }$ are reduced states of two pure states in a larger system. Assume that we have access to this larger system. Then, the distinguishability of $\hat{\rho }$ and $\hat{\sigma }$ is at least as high as the distinguishability of two arbitrary pure states whose overlap (modulus of the scalar product) is given by the fidelity $F(\hat{\rho },\hat{\sigma })$.* This interpretation assumes operation on the larger system where both $\hat{\rho }$ and $\hat{\sigma }$ are pure. Various theorems contain the sharp value of fidelity, but their meaning should always include operations on the ‘environmental’ quantum system.

Fidelities are related to distances, also called statistical distances, on the underlying Riemannian geometries. The infinitesimal distance dℓ defines the Fisher metrics [7] between the probability distributions *p* and p+dp and the Bures metrics [2] between the quantum states $d\hat{\rho }$ and $\hat{\rho }+d\hat{\rho }$: (3)$$\begin{array}{ccc} {\left(d\ell \right)}_{Fisher}^{2} & = & 2\sum_{\alpha }{\left(d\sqrt{p_{\alpha }}\right)}^{2}=\sum_{\alpha }\frac{{\left(dp_{\alpha }\right)}^{2}}{p_{\alpha }}, \end{array}$$(4)$$\begin{array}{ccc} {\left(d\ell \right)}_{Bures}^{2} & = & 2\mathrm{tr}{\left(d\sqrt{\hat{\rho }}\right)}^{2}=\mathrm{tr}\left(d\hat{\rho }\frac{2}{{\hat{\rho }}_{L}+{\hat{\rho }}_{R}}d\hat{\rho }\right). \end{array}$$
The labels L/R mean ${\hat{\rho }}_{L}$ acts from the left, as usual, while ${\hat{\rho }}_{R}$ acts from the right. The local Fisher metric is the special case of the local Bures metrics when $[\hat{\rho },d\hat{\rho }]=0$. Consider the geodesics between *p* and *q* or between $\hat{\rho }$ and $\hat{\sigma }$. The geodesic distances *ℓ* are functions of the respective fidelities: (5)$$\begin{array}{ccc} \ell {\left(p,q\right)}_{Fisher} & = & 2\arccos F(p,q)\;\in [0,\pi ], \end{array}$$(6)$$\begin{array}{ccc} \ell {\left(\hat{\rho },\hat{\sigma }\right)}_{Bures} & = & 2\sqrt{1-F^{2}\left(\hat{\rho },\hat{\sigma }\right)}\;\in \left[0,2\right]. \end{array}$$
As we said, when $\hat{\rho }$ and $\hat{\sigma }$ commute, F(q,p) is the special case of $F(\hat{\rho },\hat{\sigma })$ and the local Fisher geometry (Equation 3) is the special case of the local Bures geometry (Equation 4). This is no longer the case with the above global distances, since $\ell_{Bures}=2\sin\left(\frac{1}{2}\ell_{Fisher}\right)$. The Bures geodesic distance is shorter than the Fisher geodesic distance. The latter is the minimum length of paths but through the commuting states.

The infinitesimal distances, i.e., the local metrics, have exact meaning in metrology via the famous Cramér-Rao [8,9] and quantum Cramér-Rao theorems [10]. *Suppose an unknown state—or probability distributions p in the classical case—at an unknown length ℓ measured from one end of a known path. We perform N independent measurements on the unknown state to estimate ℓ, i.e., to estimate the unknown state. If N→∞, optimum measurements ensure that the mean-squared error of estimation goes to zero as ${\left(\Delta \ell \right)}^{2}=1/N$.* This interpretation concerns the local geometry at one point and does not tell us anything about non-local features like generic Fisher–Bures distances.

While in metrology and informatics, the natural quantum generalization of the Fisher metrics is the Bures metric, in irreversible processes, it is the Kubo–Mori–Bogoliubov (KMB) metrics (cf., e.g., ref. [11]):(7)$${\left(d\ell \right)}_{KMB}^{2}=\mathrm{tr}\left(d\hat{\rho }d\;\log\hat{\rho }\right)=\mathrm{tr}\left(d\hat{\rho }\frac{\log{\hat{\rho }}_{L}-\log{\hat{\rho }}_{R}}{{\hat{\rho }}_{L}-{\hat{\rho }}_{R}}d\hat{\rho }\right).$$
Unlike the expression of geodesic distance in terms of fidelity in Fisher–Bures metrics, no such closed expression is known in the Fisher–KMB geometry. Both geometries have been missing their global interpretations, but the close relationship of the KMB metrics with von Neumann entropies will help us. The present work proposes the operational meaning of path lengths in the Fisher–KMB metrics, which could determine the meaning of the geodesics. In the special case, we obtain the operational meaning of the classical Bhattacharyya fidelity as well. The concept is that in near-reversible transport along a path γ between two predefined states $\hat{\rho },\hat{\sigma }$, the minimum entropy production is quantified by the length $\ell_{\gamma }\left(\hat{\rho },\hat{\sigma }\right)$.

## 2. State Transport by Equilibrating Reservoirs

In preparation, we define the transport of the system’s initial state $\hat{\rho }$ into the final state $\hat{\sigma }$ in contact with a single reservoir. We use the reservoir model of ref. [12] and the theorem therein (cf. [13] for the rigorous proof).

The reservoir consists initially of *n* independent systems each in the state $\hat{\sigma }$, and we start from the composite system–reservoir state $\hat{\rho }⊗{\hat{\sigma }}^{⊗n}$.The contact of the system and the reservoir represents a reversible step, a swap between the system’s state $\hat{\rho }$ and one (e.g., the first) of the reservoir’s states $\hat{\sigma }$:(8)$$\hat{\rho }⊗{\hat{\sigma }}^{⊗n}\Rightarrow \hat{\sigma }⊗\hat{\rho }⊗{\hat{\sigma }}^{⊗(n-1)}.$$
The second step, concerning the reservoir only, is the irreversible relaxation of the reservoir to a homogeneous state, modeled by the following twirl over the permutation group:(9)$$\hat{\rho }⊗{\hat{\sigma }}^{⊗(n-1)}\Rightarrow \frac{1}{n}\sum_{k=0}^{n-1}{\hat{\sigma }}^{⊗k}⊗\hat{\rho }⊗{\hat{\sigma }}^{⊗(n-k-1)}.$$
According to this model, the transport $\hat{\rho }\to \hat{\sigma }$ of the system state leads to irreversible relaxation inside the reservoir. In the infinite reservoir limit n→∞, fortunately, the entropy production of relaxation can be expressed as the relative entropy $S\left(\hat{\rho }∥\hat{\sigma }\right)=\mathrm{tr}\left(\hat{\rho }\ln\hat{\rho }-\hat{\rho }\ln\hat{\sigma }\right)$:(10)$$\begin{array}{ccc} & & \;\;\;\;\;S\;\left(\frac{1}{n}\sum_{k=0}^{n-1}{\hat{\sigma }}^{⊗k}⊗\hat{\rho }⊗{\hat{\sigma }}^{⊗(n-k-1)}\right)-S\;\left(\hat{\rho }⊗{\hat{\sigma }}^{⊗(n-1)}\right)\\ & & \;\;\;\;\;\;\;\;\underset{n\to \infty }{\to }S\left(\hat{\rho }∥\hat{\sigma }\right). \end{array}$$
This is the entropy production of the single-step state transport as well:(11)$$\Delta S=S(\hat{\rho }∥\hat{\sigma }).$$

The entropy production can be smaller and can even go to zero if we apply sequential equilibration with intermediate reservoirs along a path, such that the step sizes go to zero. The following feature of relative entropy will be important [14]:(12)$$S\left(\hat{\rho }|\hat{\rho }+d\hat{\rho }\right)=\frac{1}{2}{\left(d\ell \right)}^{2},$$
where dℓ is defined by the Fisher metrics (Equation 3) in the classical limit, and in the general quantum case, it corresponds to the Fisher–KMB (Equation 7) and not to the Fisher–Bures (Equation 4) metrics. Consider a smooth (not necessarily geodesic) path γ from $\hat{\rho }$ to $\hat{\sigma }$. We use *N* reservoirs to transport $\hat{\rho }$ into $\hat{\sigma }$ in *N* steps. Consider a monotone sequence of intermediate states ${\hat{\rho }}_{i}$ along the path, such that $\hat{\rho }={\hat{\rho }}_{0}$ and ${\hat{\rho }}_{N}=\hat{\sigma }$. Perform the transportation $\hat{\rho }\Rightarrow {\hat{\rho }}_{1}\Rightarrow {\hat{\rho }}_{2}\Rightarrow \cdots \Rightarrow {\hat{\rho }}_{N-1}\Rightarrow \hat{\sigma }$. The total entropy production is the sum of the *N* yields:(13)$$\Delta S=\sum_{i=0}^{N-1}S\left({\hat{\rho }}_{i}∥{\hat{\rho }}_{i+1}\right).$$
In the limit N→∞, each step $\Delta \ell_{i}$ between ${\hat{\rho }}_{i}$ and ${\hat{\rho }}_{i+1}$ can go to zero. In this asymptotic regime, applying Equation (Equation 12), we can thus write
(14)$$\Delta S=\frac{1}{2}\sum_{i=0}^{N-1}{\left(\Delta \ell_{i}\right)}^{2}.$$
This is an important expression, but the r.h.s. is not yet the function of the total path length $\ell_{\gamma }\left(\hat{\rho },\hat{\sigma }\right)$, while the length’s elements are constrained by it:(15)$$\sum_{i=0}^{N-1}\Delta \ell_{i}=\ell_{\gamma }\left(\hat{\rho },\hat{\sigma }\right).$$
Keeping this constraint, let us minimize the entropy production (Equation 14). It is trivial that each term on the r.h.s. must have the same value, i.e., $\Delta \ell_{i}=\ell_{\gamma }\left(\hat{\rho },\hat{\sigma }\right)/N$ for all *i*. This means that the dense sequence ${\hat{\rho }}_{i}$ must be evenly distributed along the path, and the entropy production will be evenly distributed over the steps. The minimum of the total entropy production along the given path γ takes this form:(16)$$\Delta S_{\gamma }=\frac{\ell_{\gamma }^{2}\left(\hat{\rho },\hat{\sigma }\right)}{2N}.$$
The fastest approach to reversible transport is achieved on geodesics. In the classical case, inserting the geodesic length (Equation 5) in (Equation 16), the ultimate lower bound on entropy production reads
(17)$$\Delta S=\frac{2}{N}{\left(\arccos F\right)}^{2}.$$
This expresses the desired sharp operational meaning of the classical Bhattacharyya fidelity (Equation 1). The bounds are attainable in the limit N→∞ by operations on the system plus the reservoirs.

We find the simple meaning of the Fisher–KMB length if we introduce the number ν of equilibrations per unit length, i.e., the density $\nu =N/\ell_{\gamma }$. Then, the entropy production becomes the following linear function of the path length:(18)$$\Delta S_{\gamma }\left(\ell \right)=\frac{\ell_{\gamma }}{2\nu }.$$
This suggests the following operational meaning of the Fisher–KMB length. *Consider the near-reversible state transport via sequential equilibrations along a smooth path. If ν≫1 is the density of equilibrations per unit path length, then the entropy production is (1/2ν) per unit length.*

## 3. Remarks and Summary

Paths in Fisher geometries are perhaps the simplest non-local objects and, as such, they can be interpreted as state transport processes. It therefore seemed logical here to look for the meaning of the geometries. One should note the standard Kantorovich–Wasserstein theory [15,16], where the transport from distribution *p* to *q* happens via direct relocation of populations from $p_{\alpha }$ into $q_{\beta }$. The transport of quantum state $\hat{\rho }$ into $\hat{\sigma }$, too, can work similarly after a single unitary rotation of $\hat{\rho }$ that makes it commute with $\hat{\sigma }$. This may seem much simpler than sequential equilibration. But the simple method assumes detailed control of the states in question. If it is not possible because, e.g., the states are complicated many-body states, then the method of equilibration becomes more valuable.

Thermodynamic systems are typical examples. The Weinhold–Ruppeiner geometry [17,18] defines the thermodynamic distances between thermodynamic states and thermodynamic path lengths in general. It has long been known that thermodynamic lengths coincide with statistical lengths [19] and they are rooted in the Fisher geometry of Gibbs states [20]. In thermodynamics, the expression ${\left(\Delta \ell \right)}^{2}/V$ quantifies how the statistical fluctuations in the thermodynamic parameters tend to zero in the thermodynamic limit of infinite volume V→∞. This does not interpret the global thermodynamic lengths *ℓ* but the local metrics. Studies of thermodynamic state transport [21,22,23,24] have investigated the relationship between thermodynamic path length and thermodynamic entropy production. Salamon and Nulton [21] recognized that in discrete sequential equilibration, the equilibration rates are canceled due to the expression of entropy production, and they found the thermodynamic predecessor of Equation (Equation 16). Scandi and Perernau-Llobet derived similar results in quantum thermodynamics using the Fisher–KMB geometry of quantum Gibbs states [25]. Ref. [23] was the first attempt at constructing an underlying classical microscopic mechanism of specific irreversible state transport. The present work constructed an abstract microscopic generalization valid in the Fisher–KMB geometries of arbitrary quantum states, including the Fisher geometry special case of probability distributions. For this construction, the analytically tractable microscopic reservoir model and the related theorem, both proposed in ref. [12], were instrumental.

The global Fisher statistical distance, both in the space of probability distributions and in the space of density matrices with the KMB metrics, underwent operational interpretation for the first time. We obtained the sharp, not just qualitative, operational (physical) meaning of the classical Bhattacharyya fidelity. (Similar interpretation of the quantum fidelity and the global Fisher–Bures geometry can be the subject of future study). The concrete choice of the reservoir and the equilibration protocol are not likely to be critical, and alternative microscopic models should lead to the same relationships between entropy production and the Fisher–KMB geometry. Future investigations may replace our stepwise transport with a continuous protocol.

## Data Availability

The data presented in this study are openly available in e-print arXiv, https://doi.org/10.48550/arXiv.2410.04307, reference number arXiv:2410.04307.

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
