# Peer review of "Operational Meaning of Classical Fidelity and Path Length in Kubo–Mori–Bogoliubov Fisher Geometry"

_entropy, 2025, doi:10.3390/e27010042_

Round 1

Reviewer 1 Report

Comments and Suggestions for Authors

I really like the idea underlying the work. However, there is a major mistake that falsifies the results.

Specifically, equation 11 in the manuscript is false. The Riemannian metric tensor induced by the von Neumann-Umegaki relative entropy is not the Bures-Helstrom metric tensor (as claimed in the work), but it is the Bogoliubov-Kubo-Mori (BKM) metric tensor. A direct proof can be found here: https://doi.org/10.1063/1.530611, as well as in other works reviewing Petz's quantum monotone metric tensors in quantum information geometry. Unfortunately, unlike the case of the Bures-Helstrom metric tensor, and as far as I know, the closed form of the geodesic distance of an arbitrary geodesic of the BKM metric is not known.

In principle, this mistake is not fixable. However, I really like the idea of giving an operational interpretation to the classical fidelity through quantum considerations, so I would really like to give the author the opportunity to suitably rethink the idea in order to overcome/avoid the above-mentioned mistake.

In conclusion, I suggest a major revision.

Author Response

QUOTE: I really like the idea underlying the work. However, there is a major mistake that falsifies the results.

Specifically, equation 11 in the manuscript is false. The Riemannian metric tensor induced by the von Neumann-Umegaki relative entropy is not the Bures-Helstrom metric tensor (as claimed in the work), but it is the Bogoliubov-Kubo-Mori (BKM) metric tensor. A direct proof can be found here: https://doi.org/10.1063/1.530611, as well as in other works reviewing Petz's quantum monotone metric tensors in quantum information geometry. Unfortunately, unlike the case of the Bures-Helstrom metric tensor, and as far as I know, the closed form of the geodesic distance of an arbitrary geodesic of the BKM metric is not known.

ANSWER:  I fully agree and am seriously indebted to the criticism. (Btw, I have failed to relocate the false statement in the literature/internet that had misled me, I should have had checked it anyway.) 

QUOTE: In principle, this mistake is not fixable. However, I really like the idea of giving an operational interpretation to the classical fidelity through quantum considerations, so I would really like to give the author the opportunity to suitably rethink the idea in order to overcome/avoid the above-mentioned mistake.

ANSWER: Yes, the mistake is quite fatal. On the other hand, the whole concept, proofs, conclusions remain valid for the KMB metric. Accordingly, the revised ms retains structure and 90% of the material of the old ms.  Beyond the Bures--->KMB replacements, the Title and the Abstract are revised, lines 47-56 p2 introduce KMB metrics, lines 76-77 p3 correct the false statement of the old ms about its eq. (11) [currently (12)]. Lines 125-126 p4 refer to a quantum thermodynamic work using the KMB metric. Lines 133-137 p4 contain a revised part of the conclusion.  There are new refs. [11,14,25].  I appreciate the reviewer's insight that the underlying idea might be deserving, and I am grateful for the modest encouragement. (When the editorial process is completed, I would be pleased to name him/her in the Acknowledgement.) 

Reviewer 2 Report

Comments and Suggestions for Authors

Geometric properties of quantum dynamics provide a significant and important insight into quantum information related in particular to quantum communication processing. The paper under consideration is sound and well written. It gives an interesting and novel perspective for quantum state transport properties qualified in both informational and geometric terms. Mathematical precision and soundness equipped operational meaning of studied terms with an additional flavour. I strongly recommend publication of the paper in its present form by Entropy

Author Response

QUOTE: Geometric properties of quantum dynamics provide a significant and important insight into quantum information related in particular to quantum communication processing. The paper under consideration is sound and well written. It gives an interesting and novel perspective for quantum state transport properties qualified in both informational and geometric terms. Mathematical precision and soundness equipped operational meaning of studied terms with an additional flavour. I strongly recommend publication of the paper in its present form by Entropy

ANSWER: I am grateful for the positive assessment. An other referee remarked that I overlooked something, i.e., the eq. (11) does not yield the Bures but the KMB metrics. I had to reshape the ms for the latter.  The whole concept, proofs, conclusions remain valid for the KMB metric. The revised ms retains structure and 90% of the material of the old ms.  Beyond the Bures--->KMB replacements, the Title and the Abstract are revised, lines 47-56 p2 introduce KMB metrics, lines 76-77 p3 correct the false statement of the old ms about its eq. (11) [currently (12)]. Lines 125-126 p4 refer to a quantum thermodynamic work using the KMB metric. Lines 133-137 p4 contain a revised part of the conclusion.  There are new refs. [11,14,25].  I am sorry for the hidden problem of the old ms, and I hope you keep your trust and positive decision about the revised ms as well.